# Comparing Properties of Variable Pore-Sized 3D-Printed PLA Membrane with Conventional PLA Membrane for Guided Bone/Tissue Regeneration

**DOI:** 10.3390/ma12101718

**Published:** 2019-05-27

**Authors:** Hao Yang Zhang, Heng Bo Jiang, Jeong-Hyun Ryu, Hyojin Kang, Kwang-Mahn Kim, Jae-Sung Kwon

**Affiliations:** 1Department and Research Institute of Dental Biomaterials and Bioengineering, Yonsei University College of Dentistry, Seoul 03722, Korea; DRACOZHANG@yuhs.ac (H.Y.Z.); sweetden@naver.com (J.-H.R.); khj5797@naver.com (H.K.); 2The CONVERSATIONALIST Club, School of Stomatology, Shandong First Medical University & Shandong Academy of Medical Sciences, Tai’an 271016, Shandong, China; hengbojiang@foxmail.com; 3BK21 PLUS Project, Yonsei University College of Dentistry, Seoul 03722, Korea

**Keywords:** 3D printing, polylactide, guided bone regeneration, guided tissue regeneration, artificial membranes, biomaterials

## Abstract

The aim of this study was to fabricate bioresorbable polylactide (PLA) membranes by 3D printing and compare their properties to those of the membranes fabricated by the conventional method and compare the effect of different pore sizes on the properties of the 3D-printed membranes. PLA membranes with three different pore sizes (large pore-479 μm, small pore-273 μm, and no pore) were 3D printed, and membranes fabricated using the conventional solvent casting method were used as the control group. Scanning electron microscopy (SEM) and micro-computed tomography (µ-CT) were taken to observe the morphology and obtain the porosity of the four groups. A tensile test was performed to compare the tensile strength, elastic modulus, and elongation at break of the membranes. Preosteoblast cells were cultured on the membranes for 1, 3 and 7 days, followed by a WST assay and SEM, to examine the cell proliferation on different groups. As a result, the 3D-printed membranes showed superior mechanical properties to those of the solvent cast membranes, and the 3D-printed membranes exhibited different advantageous mechanical properties depending on the different pore sizes. The various fabrication methods and pore sizes did not have significantly different effects on cell growth. It is proven that 3D printing is a promising method for the fabrication of customized barrier membranes used in GBR/GTR.

## 1. Introduction

Guided bone/tissue regenerations (GBR/GTRs) have been widely used as treatments for alveolar bone augmentation and periodontal defects. GBR/GTR involves the use of a barrier membrane to impede the ingrowth of soft tissue into defect sites and promote periodontal tissue or bone regeneration. GBR/GTR membranes are typically classified as either bioresorbable or non-bioresorbable membranes. Bioresorbable membranes are most commonly used because they do not require secondary surgery to extract the membrane after bone healing.

Polylactide (PLA) membranes are representative bioresorbable membranes and have been commercialized for a long time [1,2]. PLA exhibits various advantageous features, including low degradation rate [3], safe degradation products (carbon dioxide and water) [4], good biocompatibility, good processability [5], and controllable permeability [6]. Commercial PLA membranes degrade within 1 year, which is ideal for complete bone growth and have also been used in various medical applications [7].

Conventional methods for barrier membrane fabrication are complex [8]. Besides, in commercialization, membranes can only be fabricated in the same shapes and if they possess identical properties, because of the constraints imposed by mass production. Therefore, individualized treatment options cannot be provided.

The technique of 3D printing has been recently introduced in medical fields. Using a desktop 3D printer, barrier membranes can be fabricated in different forms with various materials, enabling the production of customized devices for the individual condition of the patient [9]. In addition, 3D printing can be used to easily control the inner structure of the membrane and develop a distinct porous structure. In comparison, conventional methods for fabricating porous membranes involve complex procedures (e.g., solvent casting/salt leaching), where the size, form, and distribution of pores cannot be precisely controlled.

Porosity is an important feature of membranes as they allow for the infiltration of nutrients into the defect, which promotes bone growth; however, excessively large pores make the membranes less effective as a barrier against soft tissue cells [2,10,11]. The pore size of barrier membranes has remained a controversial issue since a long time. Different studies have concluded various pore sizes as optimal, and no optimal membrane pore size has been confirmed to date. The pore sizes of commercial GBR/GTR membranes range from 0.2 to 1700 μm [2,10].

No previous study has compared 3D printed and conventionally fabricated PLA barrier membranes. In addition, no comparison of mechanical properties, and cell growing ability between PLA barrier membranes with different pore sizes has been reported. The goal of this study was to fabricate PLA membranes by 3D printing and compare their properties to those of the membranes fabricated by the conventional method and compare the effect of different pore sizes on the properties of the 3D-printed membranes.

## 2. Materials and Methods

### 2.1. Specimen Fabrication

The membranes for experimental groups were prepared by fuse deposition modeling (FDM), a common method for 3D printing. A model of the desired specimen was created using a modeling software (123D Design, Autodesk, CA, USA) with a thickness of 0.4 mm. The exported STL file was loaded in a slicing software program (Cubicreator3, HYVISION SYSTEM, Seongnam-si, Korea). The infill rate was set at 60%, 80% or 100% for the production of membranes with different pore sizes (Figure 1a–c). The exported G-code file was then introduced into a 3D printer (CUBICON 3DP-310F, HYVISION SYSTEM, Seongnam-si, Korea), which used PLA filament from the same manufacturer for printing in three layers.

The printed specimen was observed using an image analyzer and its pore size was measured. The average pore sizes were as follows: 60% infill rate: 479 μm, 80% infill rate: 273 μm, 100% infill rate: no observed pores. Each test group was assigned a code: LP (large pore), SP (small pore), and NP (no pore).

The membranes for the control group were prepared by solvent casting, which is a conventional method for membrane fabrication. The filament was dissolved in N, N-dimethylformamide (Sigma-Aldrich, St. Louis, MO, USA) at a 5% mass/volume concentration. The solution was poured into a mold and placed in a drying oven at 100 °C for 6 h for evaporation. The completely dried membrane was obtained and cut into the desired forms. The thickness of the membranes were kept the same as the 3D-printed groups. The membranes prepared using the solvent casting were coded SC.

### 2.2. Scanning Electron Microscopy

Membranes were coated with Au in a sputter coater and the morphology of the membranes was observed by scanning electron microscopy (SEM, Jeol JSM-6700F, JEOL, Akishima, Japan) at an accelerating voltage of 10 kV.

### 2.3. Micro-Computed Tomography

The membranes of each group were scanned by micro-computed tomography (µ-CT, SkyScan1076, Bruker, Billerica, MA, USA). The µ-CT scanner settings were specified to be 100 kV, 100 μA, and 360° rotation with a pixel size of 36 μm. The 3D images of membranes were obtained by 3D reconstruction and the porosities of membranes were calculated from their respective µ-CT images using the 3D analysis program from the manufacturer (CTAn 1.12.0.0, Bruker, Billerica, MA, USA).

### 2.4. Tensile Test

The tensile tests were performed using a universal testing machine (Instron 5942, Instron, Norwood, MA, USA) with specimen dimensions of 150 × 10 mm^2^ (Figure 1d) and an initial distance between grips of 100 mm. A tensile force was applied at a crosshead speed of 5 mm/min until the specimen broke. The tensile strength and elastic modulus were recorded in units of MPa, and the elongation at break was recorded as a percentage value.

### 2.5. Cell Proliferation Test

The diameter of the circular specimen for the cell proliferation test was 10 mm (Figure 1e). Murine preosteoblast cell line MC3T3-E1 and culture medium consisting of Alpha Minimum Essential Medium (α-MEM) supplemented with 10% fetal bovine serum and 1% antibiotic/antimycotic (GIBCO, Waltham, MA, USA) were used. The membranes were sterilized under UV overnight and placed into 12-well culture plates under aseptic conditions. Cell suspensions of 1 × 10^4^ cells in 100 μL of the above-described medium were dripped on each membrane and incubated under 5% CO_2_ and 95% relative humidity at 37 °C. After 4 h, 1 mL of the medium was added to each well, submerging the membrane, and incubating under same conditions. The medium was changed every other day.

After incubation for 1, 3, and 7 days, membranes were placed in new plates and medium and 100 μL of WST solution were added to each well and incubated for 2 h. Then, 100 μL of the solution in each well was transferred to a 96-well plate. Absorbance was determined using a microplate spectrophotometer (Epoch, BioTek, Winooski, VT, USA) at 450 nm.

The cells on the membranes were observed with SEM (S-3000N, Hitachi, Japan). After fixation with 2% glutaraldehyde, 2% paraformaldehyde, and 0.5% calcium chloride for 6 h, the samples were washed with 0.1 M PBS for 1 h, fixed with 1% osmium tetroxide for 2 h and washed for 30 min, followed by dehydration with graded ethanol solutions (50%, 60%, 70%, 80%, 90%, 95%, 100% and 100%). The resultant samples were freeze-dried and coated with platinum using an ion coater (E-1010, Hitachi, Tokyo, Japan) for SEM observation at an accelerating voltage of 15 kV.

### 2.6. Statistical Analysis

All data are expressed as means ± standard deviation. Statistical analysis was performed using one-way analysis of variance (ANOVA) followed by a Tukey’s post hoc test and *p* < 0.05 were considered significant.

## 3. Results

### 3.1. General Characterization

Figure 1f–i shows the SEM images of the four groups where the printed struts appear to be perpendicularly crossed and coarse surfaces were observed on the printed membranes. The width of the printed struts was not consistent. For the LP group, the average width of the struts was 278 ± 58 μm and average distance between struts was 397 ± 48 μm. For the SP group, the average width of struts was 264 ± 49 μm and the average distance between them was 188 ± 43 μm. For the NP group, the average width of struts was 380 ± 27 μm and no space was observed between struts except for some defects. The SC group showed a relatively smooth surface, but a small amount of shallow pits was observed.

Figure 2 shows the reconstructed µ-CT image of the four groups. Similar to SEM, the widths of the struts and pore sizes were not consistent in all sections of the membranes. A few grooves but no obvious pores were seen on the surface of the NP membrane (Figure 2c). Pits were observed in the µ-CT image of the SC membrane (Figure 2d) and Table 1 shows the porosity of the four groups as determined by µ-CT. The porosities of the NP and SC groups were similar and small, which was likely caused by occasional defects, as seen in the SEM and µ-CT images. The porosity of the SP was significantly higher than those of NP and SC, and LP had the significantly highest porosity.

### 3.2. Tensile Test Results

Figure 3 shows representative stress-strain curves of the tensile tests of the prepared membranes, and Figure 4 shows the tensile test results of the four groups. In terms of tensile strength (Figure 4a), the NP group showed the highest value followed by the SC group. The tensile strengths decreased as the pore size increased from NP to SP to LP. Statistically significant differences were observed between all groups tested. In terms of the elastic modulus (Figure 4b), the SC group was the highest followed by the NP group. The elastic moduli also decreased as the pore size increased. Statistical significance was observed between the values of all groups. In terms of elongation at break (Figure 4c), the LP group showed significantly higher values than all other groups, followed by the SP and NP, while the SC showed the significantly lowest value.

### 3.3. Cell Proliferation Test Results

Figure 5 shows the absorbance values measured in the WST test after 1, 3, and 7 days of cell proliferation on the prepared membranes. On day 1, the LP group exhibited a significantly lower absorbance than those of the NP and SC groups. On day 3, all groups showed increased absorbance compared to their respective measurements taken on day 1, and the LP group absorbance was significantly lower than that of the SC group. On day 7, all groups showed a drastic increase in absorbance and no statistically significant difference was observed between groups.

Figure 6 shows the SEM images of the MC3T3-E1 preosteoblast cells cultured on LP, SP, and NP membranes for 1 and 3 days. The number of cells increased in all three groups from day 1 to 3. The cells became so dense on day 3 that some areas already exhibited cell detachment, and cell density was similar in all three groups. In the NP group, cells mainly grew on the outer surface of the membrane, whereas in the LP and SP groups, cells grew on the strut edges into the inside of the membrane and onto the surface of the inner struts.

## 4. Discussion

The use of barrier membranes in tissue regeneration was first introduced in 1959 [12]. The studies and applications of barrier membranes over the past few decades have revealed a great deal of insight into this method. According to Scantlebury [13], barrier membranes must fulfill five essential design criteria: tissue integration, cell-occlusivity, clinical manageability, spacemaking, and biocompatibility. These characteristics are governed by the composition, physicochemical properties, and structure of the membrane.

The use of 3D printing for fabricating medical implants is still in the development stage. However, this fabrication method has undeniable potential, especially as individualized treatment is becoming popular for the treatment of a variety of disease states. Previously, 3D printing has been used to make custom implants in cranial surgery, dentistry, and maxillofacial surgery, leading to reduced treatment time, improved accuracy, and better medical outcomes [14]. Following this trend, it is likely that 3D printing will become one of the most effective methods for medical devices fabrication.

The 3D printing of GTR/GBR barrier membranes would be fast, convenient, and practical. Barrier membranes generally have a simple structure, but their integration with adjacent tissue is vital for the desired outcomes. When utilizing commercially available membranes, pretreatment is typically performed by manually trimming and bending the uniform-structured membranes to fit them to the various bone contours. Using this technique, complete tissue integration and membrane stabilization cannot be guaranteed and it can possibly lead to fibrous connective tissue ingrowth. Therefore, membrane customization with 3D printing is a promising alternative to conventional membrane fabrication methods [9].

Additionally, 3D printing allows for the creation of specific pore architecture, which has been deemed a critical feature of barrier membranes. Discussion regarding the optimum membrane pore size has been ongoing over the past two decades. Generally, a microporous membrane would have the following features: prevention of ingrowth of soft tissue; reduced bacterial penetration; large surface area for cell attachment; reduced integration with tissue allowing for a simple pullout for retrieval; blunt edges to prevent tissue injury in the event of breakage; as well as high strength and elastic modulus leading to better space maintenance but inferior manageability. In contrast, features of a typical macroporous membrane include good wound stability through tissue integration, but the need for surgical separation for removal. Also, macroporous membranes would have good nutrient infiltration, and high flexibility for clinical manipulation, but inferior space maintenance [2,10,11]. A few animal studies have been performed to test and compare the effect of microporous and macroporous membranes as well as porous and nonporous membranes on bone regeneration, which have drawn diverse conclusions. Marouf et al. [15] covered rabbit calvarium defects with microporous PTFE membranes and macroporous expanded PTFE membranes, and bone regeneration was examined. Greater rates and quantities of bone regeneration were observed in the defective cavities covered with the macroporous expanded PTFE membrane than those covered with the microporous PTFE membrane. Pineda et al. [16] used microporous (2–4 μm), medium (10–20 pm), and large (20–200 pm) pore PLLA membranes to implant in rabbits to cover a radius defect. The authors found that regenerating bone with the microporous membranes was the most effective, followed in order by the medium and large pore-sized membranes. Simion et al. [17] investigated the effect of an open-pore membrane and totally occlusive membrane on ridge defects in dog mandible, demonstrating that the totally occlusive membrane achieved the best regenerative capacity. In contrast, Lundgren et al. [18] used an occlusive barrier membrane and membranes with perforations ranging from 10 to 300 μm covered on rat skull, finding that the totally occlusive barrier membrane resulted in a slower rate of bone tissue augmentation. In addition, many other studies provided inconsistent conclusions similar to those described above [19,20,21,22,23,24,25]. Therefore, it can be inferred that the previous studies regarding membrane pore size did not meet a set of strict and consistent requirements. Thus, the real cause of the different conclusions provided by the respective authors cannot be determined. The previous studies were mainly performed in animals, so we attempted to performed fundamental investigations to determine the root causes of the problem. In this study, in vitro tensile tests and cell proliferation tests were performed to investigate the effect of two fabrication methods and three pore sizes on the mechanical properties of the membrane and initial cell attachment and growth.

The results of the tensile test are shown in Figure 3 and Figure 4. Generally, the tensile strength and elongation at break should be maximized so that the membrane can bear more load and deformation before breakage. In addition, the elastic modulus should fall within a proper range because an excessively large value would hamper clinical manageability and cause tissue damage, while an insufficient modulus would compromise space maintenance. According to the literature, commercially available bioresorbable membranes have tensile strengths ranging from 3.5 to 22.5 MPa, and elastic moduli ranging from 30.6 to 700 MPa [1,26,27,28]. Considering that the major disadvantage of bioresorbable membranes is their lack of space maintenance, an elastic modulus value slightly higher than this range would be ideal for the novel membranes. When comparing the NP and SC groups, the NP membranes showed higher tensile strength and elongation at break values, in addition to lower elastic moduli which were similar to those of commercial membranes. From these three parameters, it is clear that the 3D-printed membrane was superior to the solvent cast membrane. When comparing 3D-printed groups with different pore sizes, NP showed the highest strength and elastic modulus but a low elongation at break; SP exhibited an ideally moderate modulus, a moderate strength and a low elongation at break close to NP; and LP was the most ductile, but the lowest on tensile strength and elastic modulus. These results can be well expected considering that the materials usually become less “stiff” when the density decreases as porosity increases. From these results, it can be inferred that each of the different pore-sized membranes exhibited its own unique superiority, which can meet the requirements for various clinical conditions. For instance, if a stronger support is needed in applications like large bone defects or bone augmentation, NP membrane can be considered; for the applications requiring not ultimate strength but better manageability, like in small GTR defects or with the support of bone substitutes, the LP membrane is preferable; and for those cases where moderate properties are required, the SP membrane shall be tried. It can also be inferred from the results that 3D printing has the potential for fabricating different structured membranes with any pore size—rather than just three—to grant the membranes with proposed mechanical properties and fulfilling any kinds of clinical demand.

The results of the cell proliferation test are shown in Figure 5. When comparing the NP and SC groups, the cell growth was largely the same at all time points. A situation where the residual solvent inhibits cell viability did not occur in this test likely because the amount of residual solvent was almost negligible after a relatively complete evaporation. Comparing the membranes with different pore sizes, although at all time points cell proliferation increased with decreasing pore size, significant differences were only observed at day 1 and day 3. Because of the rapid and continuous growth of the cells, differences between groups with different pore sizes diminished over time. The mechanism underlying this result can be speculated upon by observing the SEM images of the cells (Figure 6). Comparing the images taken at days 1 and 3, the membranes with smaller pores provided smoother and larger outer surfaces, facilitating initial cell attachment, while the membranes with larger pores had a smaller outer surface and a larger inner wall area, allowing the cells to grow slowly into the inner structure. Therefore, considering the thinness of the barrier membranes, the specific surface areas of the membranes available for cell attachment were similar regardless of pore size. Furthermore, the cells attached in early stages are progenitor cells essential for the following bone tissue regeneration within the membrane-protected spaces. It can be speculated that the different patterns of cell attachment would affect tissue integration, where monolayer cells grown on small-pored membranes would lead to reduced tissue integration, and spatial cell growth in large-pored membranes leads to tighter tissue integration. This conclusion is in accordance with previous studies [15]. Subsequently, the smaller pore membrane would allow for an easy retrieval process, while the larger pore membranes exhibit enhanced membrane stability. However, a comparison between the membranes concerning final bone regeneration did not result in consistent conclusions in the literature. Therefore, this in vitro cell test showed that different pore sizes resulted in negligible effects on cell growth and subsequent bone regeneration. According to previous studies, the healing response is governed by a variety of factors, including the chemical composition, physical and surface characteristics, thickness, porosity, and the depth of the membrane placement in the tissue [10].

## 5. Conclusions

In conclusion, membranes fabricated by 3D printing showed improved mechanical properties compared to those produced via the conventional solvent casting method. Different pore sizes showed different advantages in terms of mechanical properties. The effects of both variables, the fabrication method and pore size-on cell growth, were insignificant. Therefore, 3D printing is proven to be a promising method for the fabrication of customized barrier membranes used in GBR/GTR. When possible, the membranes should be designed and 3D printed with specific pore sizes and materials and in specific forms depending on the clinical condition of the patient, achieving a more personalized treatment.

## Figures and Tables

**Figure 1 materials-12-01718-f001:**
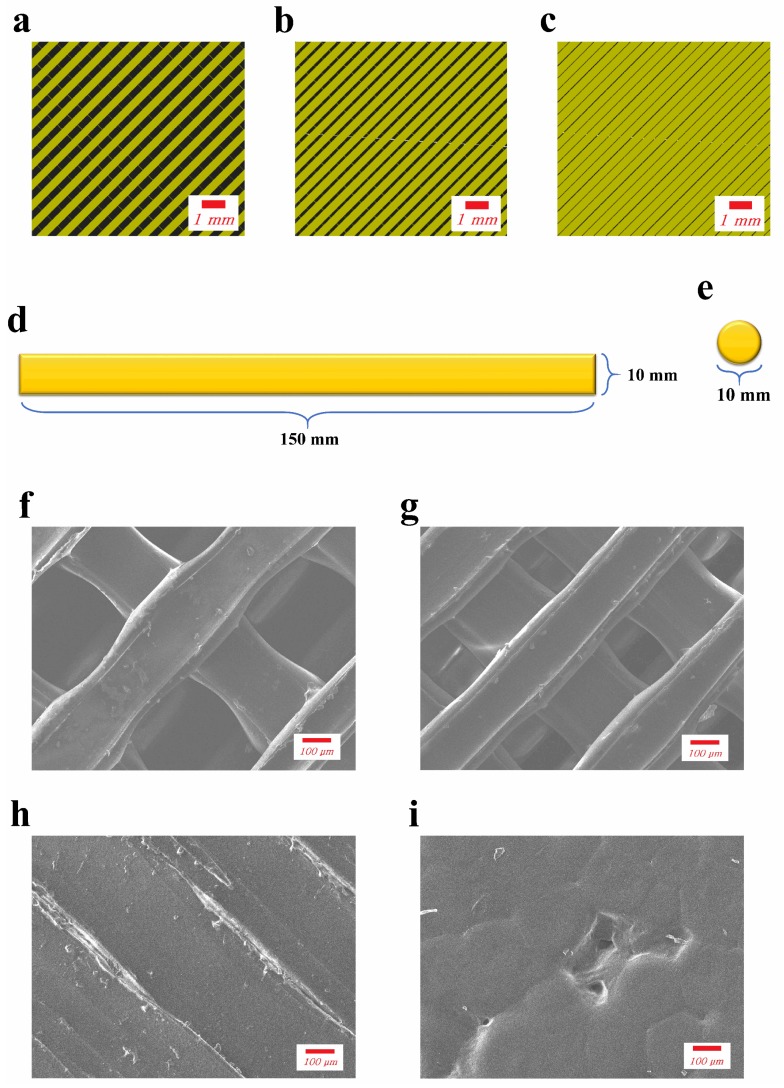
Models in slicing software program with (**a**) 60%, (**b**) 80% and (**c**) 100% infill rate; diagrams of specimens for (**d**) tensile test and (**e**) cell proliferation test; SEM images of (**f**) LP (**g**) SP (**h**) NP and (**i**) SC membranes.

**Figure 2 materials-12-01718-f002:**
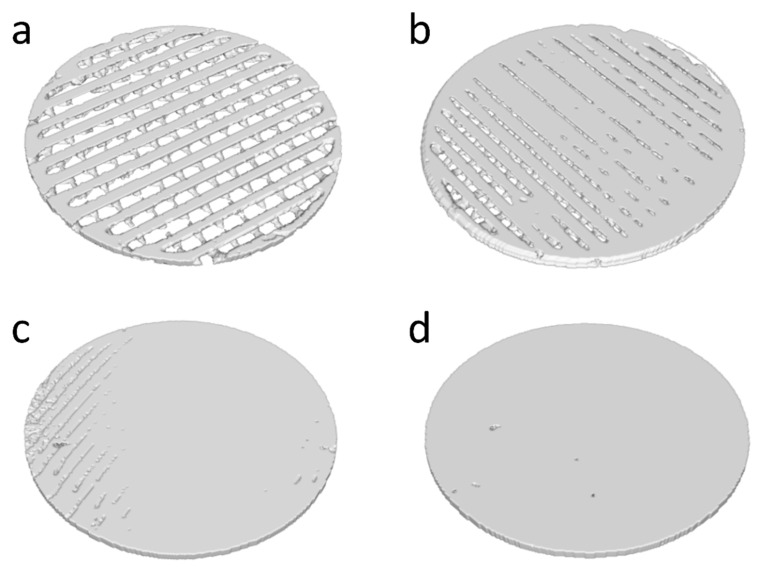
3D reconstructed µ-CT image of (**a**) LP, (**b**) SP, (**c**) NP, and (**d**) SC membranes.

**Figure 3 materials-12-01718-f003:**
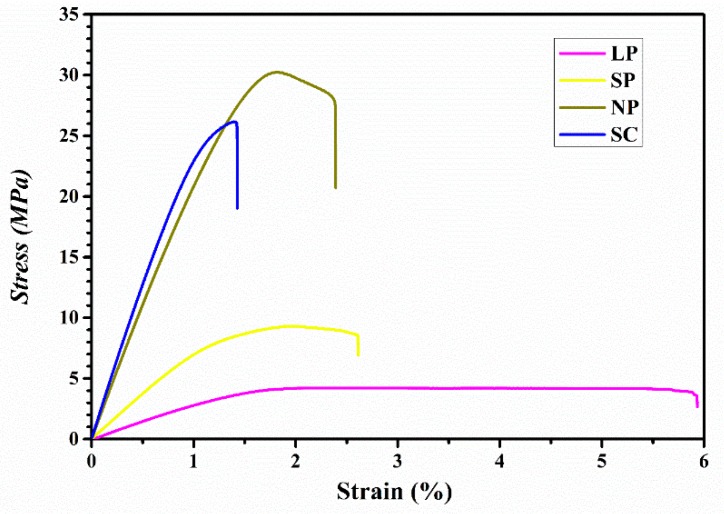
Representative stress-strain curves of the tensile tests performed on LP, SP, NP, and SC membranes.

**Figure 4 materials-12-01718-f004:**
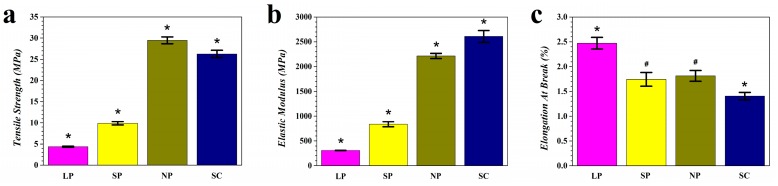
Results of the tensile test. (**a**) Tensile strength; (**b**) Elastic modulus; and (**c**) Elongation at break. (*n* = 5; * *p* < 0.05 compared to all other groups, ^#^
*p* < 0.05 compared to LP and SC groups).

**Figure 5 materials-12-01718-f005:**
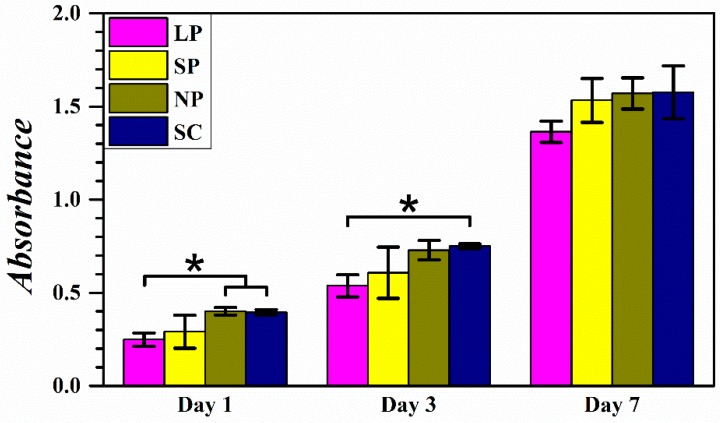
Absorbance in WST tests after 1, 3, and 7 days of cell proliferation on LP, SP, NP, and SC membranes (*n* = 3; * *p* < 0.05).

**Figure 6 materials-12-01718-f006:**
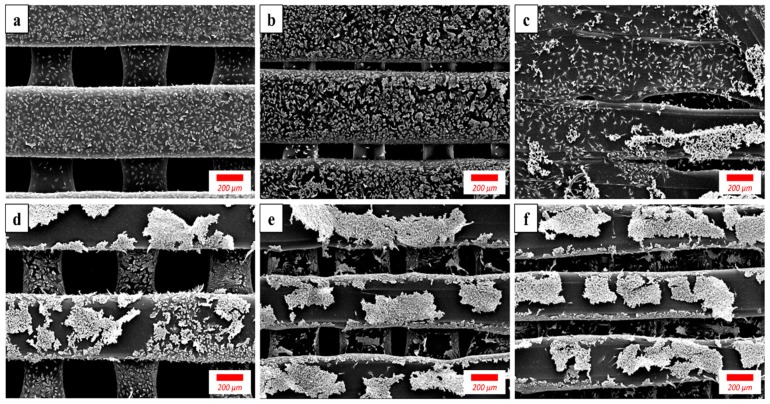
SEM images of preosteoblast cells grown on (**a**,**d**) LP, (**b**,**e**) SP and (**c**,**f**) NP membranes for (**a**–**c**) 1 day and (**d**–**f**) 3 days.

**Table 1 materials-12-01718-t001:** Porosity of LP, SP, NP, and SC membranes as determined by µ-CT.

Group	Porosity (%)
LP	51.5 ± 8.2 ^a^
SP	32.5 ± 4.9 ^b^
NP	1.3 ± 1.0 ^c^
SC	1.6 ± 2.2 ^c^

Different superscript letters indicate significant differences between the groups (*p* < 0.05).

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
