# Peer review of "Comparing Properties of Variable Pore-Sized 3D-Printed PLA Membrane with Conventional PLA Membrane for Guided Bone/Tissue Regeneration"

_materials, 2019, doi:10.3390/ma12101718_

Round 1

Reviewer 1 Report

In the present study, novel fabrication method of GBR/GTR membrane using 3D printer are presented and compared the physical properties. The reviewer feels this paper is scientifically sound but the reviewer expects to have some comments and revision by authors.

The most expected feature of GBR/GTR membrane is occlusiveness, without dehiscence of soft tissue, but this paper did not study this point. This means that proposed novel membrane is not proved to have a proper barrier property. The reviewer feels, this paper is still potentially acceptable without studying barrier property of novel membrane using animal study, but the possible occlusiveness of novel membrane should be carefully discussed.

In Introduction they indicated that no optimal membrane pore size has been confirmed to date, but this description is confusing because readers may expect after reading this sentence that this paper intends to elucidate the optimal membrane pore size. But unfortunately this study never proposed optimal membrane pore size too.

line 266-269 (the stronger NP membrane…in moderate case.): The reviewer feels it is too much farfetched to propose the application of membrane from the viewpoint of membrane strength only.

line278- (Comparing the images taken at days 1 and 3, the membranes with smaller pores provided smoother and larger outer surfaces, facilitating initial cell attachment, while the membranes with larger pores had smaller outer surface and a larger inner wall area, allowing the cells to grow slowly into the inner structure.) : The authors says the difference of surface roughness may influence the WST results. But surface areas varied among the membrane, and cells seeded onto the large-pored membrane may fall through the pore, without attaching anywhere on the membrane. Thus the analysis of cell growth rate from initially-attached cell is required. Seeded cell # was same among all groups but larger number of cells may “sieved” from the pore and initially-attached cell # must be different.

Why initial attachment of osteoblastic cells was tested? Occlusiveness and soft tissue cell attachment are felt to be further important as a GBR/GBR membrane than the attachment of osteoblast.

In Conclusion, “3D printing fabricated membranes with improved mechanical properties compared 297 to those produced via the conventional solvent casting method” Which result indicates this? Fig. 3 has no statistical comparison and in Fig. 4 no difference is indicated between SC and NP.

line 300: “3D printing can be used to replace conventional fabrication…” It may be difficult to discuss this issue without mentioning cost-effectiveness, ease of process for forming, etc.

<Minor points>

lines 109 and 110: “1x104” “4” should be superscripted; “CO2” “2”should be subscripted.

Authors name in the text: All letters are uppercased but please describe like “Smith”.

Author Response

Reviewer #1  

Major Concerns 1

The most expected feature of GBR/GTR   membrane is occlusiveness, without dehiscence of soft tissue, but this paper   did not study this point. This means that proposed novel membrane is not   proved to have a proper barrier property. The reviewer feels, this paper is   still potentially acceptable without studying barrier property of novel   membrane using animal study, but the possible occlusiveness of novel membrane   should be carefully discussed.

Response to 1

Thank you for your comments. As far as we   know, although the occlusiveness of membrane is affected by its pore size,   the relationship is not so apparent. It’s very unlikely to determine the   degree of occlusiveness directly from the value of pore size, while animal   tests are usually required. And to date there’s hardly any effective in vitro   test method to confirm the occlusiveness of membrane. Therefore, due to the   differed aim of our current research, the in vitro test focused on the   initial cell attachment and growth on the membrane, instead of the occlusiveness   of membrane.

Major Concerns 2

In   Introduction they indicated that no optimal membrane pore size has been   confirmed to date, but this description is confusing because readers may   expect after reading this sentence that this paper intends to elucidate the   optimal membrane pore size. But unfortunately this study never proposed   optimal membrane pore size too.

Response 2

Like the other researches mentioned in the   references, this research also intended to find out an optimal membrane pore   size (although by more fundamental methods). But as the results showed: 1)   the effects of different pore sizes on cell growth were not significantly   different. 2) the effects of different pore sizes on membrane’s mechanical   properties were not absolutely advantageous or disadvantageous.   Comprehensively, we drew the conclusion that there is no optimal membrane   pore size concerning this research.

Major Concerns 3

line 266-269 (the stronger NP membrane…in moderate   case.): The reviewer feels it is too much farfetched to propose the   application of membrane from the viewpoint of membrane strength only.

Response 3

We agree with your comment. We admit that this   sentence seems a bit misleading with its expression. The original purpose of   this sentence was, by relating to the required mechanical properties of   different applied ranges, to emphasize the superiority of different   pore-sized membranes, but not to point out their exact applications. We   revised the sentence with a more precise expression as the following. We hope   this could make it more rational.

Revised sentences:

For instance, if a stronger support is needed in   applications like large bone defects or bone augmentation, NP membrane can be   considered; if for the applications requiring not an ultimate strength but   better manageability, like in small GTR defects or with the support of bone   substitutes, LP membrane is preferable; and for those cases where moderate   properties are required, SP membrane shall be tried. It can also be inferred   from the results that 3D printing has the potential of fabricating different   structured membranes with any pore sizes - rather than just three - to grant   the membranes with proposed mechanical properties and fulfilling any kinds of   clinical demand.

Major Concerns 4

line278- (Comparing   the images taken at days 1 and 3, the membranes with smaller pores provided   smoother and larger outer surfaces, facilitating initial cell attachment,   while the membranes with larger pores had smaller outer surface and a larger   inner wall area, allowing the cells to grow slowly into the inner structure.)   : The authors says the difference of surface roughness may influence the WST   results. But surface areas varied among the membrane, and cells seeded onto   the large-pored membrane may fall through the pore, without attaching   anywhere on the membrane. Thus the analysis of cell growth rate from   initially-attached cell is required. Seeded cell # was same among all groups   but larger number of cells may “sieved” from the pore and initially-attached   cell # must be different.

Response 4

In fact, not only   the cell growth rate but also the penetration and non-attachment of cells   should be the factors to consider in the test. If cells would “sieved” from   the pores in this test, they also will clinically, so this kind of simulation   would be practical. In contrast, if only cell growth rate is considered, the   test would be more related to the material’s chemical composition and surface   microstructure, but not its macrostructures, so the significance of using   different pore sizes will no longer exist.

Major Concerns 5

Why initial   attachment of osteoblastic cells was tested? Occlusiveness and soft tissue   cell attachment are felt to be further important as a GBR/GBR membrane than   the attachment of osteoblast.

Response 5

Thank you for your   comment, but we think you might have mistaken the purpose of this test. This   test was aimed at observing the initial attachment and growth   of osteoblastic cells, in order to find out the real cause for the   different effects of different membrane pore sizes on the consequent bone   regeneration, because for the bone regeneration, proliferation of   osteoblastic cells is one of the most important factors. In the meanwhile,   attachment and growth of soft tissue cells is much less significant. Either   the soft tissue attach tightly on macroporous membranes or they attach weakly   on the microporous membranes are acceptable due to their respective advantage   and disadvantage (reference 10). And there’s no evidence showing soft tissue   attachment have direct influence on the occlusiveness. So this is not the   focus in our research. In previous studies, the growth of soft tissue cells   on membrane were usually tested for the cytocompatibility of the material,   which is not much related to the structure of the membrane. And as was   mentioned in response 1, occlusiveness of membrane is usually studied by   animal tests. It does not consist with the aim of our research.

Major Concerns 6

In   Conclusion, “3D printing fabricated membranes with improved mechanical   properties compared 297 to those produced via the conventional solvent   casting method” Which result indicates this? Fig. 3 has no statistical   comparison and in Fig. 4 no difference is indicated between SC and NP.

Response 6

The statistical   comparisons were revealed in the figure legend of Fig. 4 and the result section,   but not included in the figures. We apologize for any confusion. Now we have   revised the Fig. 4 by adding signs of significant differences, and revealed   the meaning of the signs in the figure legend. We also found a mistaken   statement in the results section so we revised the sentence.

Revised Fig. 4 and figure legend:

Figure. 4.   Results of the tensile test. (a) Tensile strength (b) Elastic modulus (c)   Elongation at break. (n=5; * p<0.05 compared to all other groups, #   p<0.05 compared to LP and SC groups).

Revised sentences in Tensile test results:

In terms of   elongation at break (Fig. 4c), the LP group showed significantly higher   values than all other groups, followed by the SP, NP, while the SC showed the   significantly lowest value.

Major Concerns 7

line 300: “3D   printing can be used to replace conventional fabrication…” It may be   difficult to discuss this issue without mentioning cost-effectiveness, ease   of process for forming, etc.

Response 7

Although we have   partly mentioned these issues in discussion (“Previously, 3D printing has   been used to make custom implants in cranial surgery, dentistry, and   maxillofacial surgery, leading to reduced treatment time, improved accuracy,   and better medical outcomes.”), we agree that this sentence could have been   expressed in a more precise way. Now we have revised the sentence as below:

3D printing is   proven to be a promising method for the fabrication of customized barrier   membranes used in GBR/GTR.

A corresponding   sentence in abstract have also been revised:

It is proven that   3D printing is a promising method for the fabrication of customized barrier   membranes used in GBR/GTR.

Minor points

lines 109 and 110:   “1x104” “4” should be superscripted; “CO2” “2”should be subscripted.

Authors name in the   text: All letters are uppercased but please describe like “Smith”

Response

Thank you for your   kind reminder. The mistakes have been corrected as you instructed, and author   name revised have been marked red.

Reviewer 2 Report

The authors have used 3D printing technology to design and fabricate membranes with various pore sizes and compared them to similar membranes prepared using traditional solvent casting techniques. The manuscript is well written and is substantiated by sound experiments. However, certain shortcomings, in terms of interpretation of results and experimental design need to be addressed. Here are the points that need to be addressed:

) Page 3, line 109, it should be 1x104

) In Figure 1, f-i, SEM images for selective sections are given. A broader SEM portfolio at a lower magnification must be provided so that the homogeneity in pore distribution obtained using  3D printing can be evaluated. The authors also mention that "the width of the printed struts was not consistent". 

) What is the logic behind using 60% and 80% infill rate for creating the porous membranes? Have the authors tried lower infill rates? Does a 70% infill rate provide a more consistent pore structure as compared to 80%? From the μ-CT images it is quite evident that the membranes resulting out of the 80% infill rate also have a lot of grooves, with a relatively lower number of pores as compared to the 60% infill rate.

) Did the authors use a software to calculate the porosity of the films from the micro-CT images? If yes, kindly mention that software.

) Page 8, line 200, SCANTELBURY, all caps should be modified.

) Page 8, line 211, ...GBR barrier membranes would be fast, convenient and applicable? What do the authors mean by applicable?

) Some reference for the claim that "membrane customization with 3D printing is a promising alternative to conventional membrane fabrication methods" must be provided. 

) Page 8, for line 230, MAROUF and line 240, LUNDGREN, all caps should be modified. This is a recurring mistake in the paper and should be modified.

) On page 8, the authors mention, "the previous studies regarding membrane pore size did not meet a set of strict and consistent requirements. Thus, the real cause of the different conclusions provided by the respective authors cannot be determined." However, the materials tested in each of these examples is not the same and the properties observed could be explained by the difference in material's composition.

) The authors mention on page 9, "When comparing the NP and SC groups, the NP membranes showed higher tensile strength and elongation at break values, in addition to lower elastic moduli which were similar to those of commercial membranes." This is not correct. The various properties, as clearly evident from Figure 4 show comparable values for NP and SC groups. In fact, the NP groups with pores (LP and SP) show lower values for all the three properties. Hence mentioning that the 3D printed NP group works has superior properties is not correct and must be modified. Also a clear trend in the various properties is observed as we move from the LP, SP to NP formulations. It seems quite evident that as the pore size decreases, the tensile strength and elastic modulus increase, suggesting a firmer underlying structure. These observations must be explained and related to the structural integrity and overall morphology of the membrane.

) On page 9, the authors mention "When comparing the NP and SC groups, the cell growth was largely the same at all time points...". However, in Figure 6, no images for the SC group membranes are provided. Kindly provide those SEM images for comparison.

) Sentence restructuring in the conclusion section. Verb and tense corrections in the text.

) Ideally, the membranes made using 3D printing technology, especially the ones with pores, must be compared with similar porous membranes made using traditional solvent casting (SC) techniques. I would recommend adding some control experiments with porous membranes made using SC or if that is not possible then citing data from previously published research to draw out conclusions, especially in terms of the elasticity related properties of these membranes.

Author Response

Reviewer #2  

Major Concerns 1

Page   3, line 109, it should be 1x104

Response to 1

Thank you for your kind reminder. The   mistakes have been corrected.

Major Concerns 2

In   Figure 1, f-i, SEM images for selective sections are given. A broader SEM   portfolio at a lower magnification must be provided so that the homogeneity   in pore distribution obtained using  3D   printing can be evaluated. The authors also mention that "the width of   the printed struts was not consistent".

Response 2

Thank   you for your comment. We consider that including SEM images with lower   magnification would be possible but not really necessary. The widths of   printed struts and their inconsistency can already be observed in the   existing SEM images. Besides, the main purpose of SEM was to observe the microstructures   of the membranes, and that’s why we also included the μ-CT as a supplementary   method for the macrostructural observation. We believe that the combination   of these two kinds of images should be enough for revealing various   structural features of the membranes. And we also concern that excessive   similar figures may take too much space and make the paper tedious. So we   kindly ask you to reconsider this suggestion.

Major Concerns 3

What is the logic behind using 60% and 80% infill   rate for creating the porous membranes? Have the authors tried lower infill   rates? Does a 70% infill rate provide a more consistent pore structure as   compared to 80%? From the μ-CT images it is quite evident that the membranes   resulting out of the 80% infill rate also have a lot of grooves, with a   relatively lower number of pores as compared to the 60% infill rate.

Response 3

Yes, we have tried various infill rates from 10% to   100% and decided to use 60% and 80% at last. As was mentioned in the   discussion, some previously published papers were referred to, and we chose   the proper infill rates so that the tensile strengths of experimental groups were   not below the range concluded from those papers. Also, we made sure pore   sizes fall within a proper range and the properties between groups showed   enough differences.

We think that the consistency of pore structure is not   influenced by the infill rate but only determined by the accuracy of the 3D   printer. If we simply look at the struts width, we can see that samples with   60% infill rate were not more consistent than samples with 80% infill rate. Only   because the pores were smaller with 80% infill rate, some adjacent struts merged   together and pores seemed to disappear in some spots in sample with 80%   infill rate.

Major Concerns 4

Did the authors use a software to calculate the   porosity of the films from the micro-CT images? If yes, kindly mention that   software.

Response 4

Thank you for your kind reminder. Now the software   has been added in the manuscript.

Major Concerns 5

Page 8, line 200, SCANTELBURY, all caps should be   modified.

Response 5

The author names   have been modified and marked red.

Major Concerns 6

Page 8, line 211, ...GBR barrier membranes would be   fast, convenient and applicable? What do the authors mean by applicable?

Response 6

Sorry to have caused misunderstanding due to our   inappropriate expression. What we tried to express here was that the 3D   printing will have good practicability/usability in the fabrication of GTR/GBR   membranes. After careful consideration, we have revised the sentence as   below:

The 3D printing of GTR/GBR barrier membranes would   be fast, convenient, and practical.

Major Concerns 7

Some reference for the claim that "membrane   customization with 3D printing is a promising alternative to conventional   membrane fabrication methods" must be provided.

Response 7

A reference has been added.

Major Concerns 8

Page 8, for line 230, MAROUF and line 240, LUNDGREN,   all caps should be modified. This is a recurring mistake in the paper and   should be modified.

Response 8

The author names   have been modified and marked red.

Major Concerns 9

On page 8, the authors mention, "the previous   studies regarding membrane pore size did not meet a set of strict and   consistent requirements. Thus, the real cause of the different conclusions   provided by the respective authors cannot be determined." However, the   materials tested in each of these examples is not the same and the properties   observed could be explained by the difference in material's composition.

Response 9

Indeed, the materials used in examples were not all   the same, and that’s also why we said “the previous studies regarding   membrane pore size did not meet a set of strict and consistent requirements.”.   However, as have been mentioned in discussion, the influencing factors of   bone regeneration are various. So it may not be rigorous to claim that the contrary   conclusions between these studies should be explained by different materials   used. From the result of our research, we can exclude the possibility of   influence of pore sizes on initial osteoblastic cells’ attachment and growth.   As for the other influencing factors, other tests will be needed to confirm   their influences. Still, thank you for your helpful comment. We believe that   this will benefit us with our future research directions.

Major Concerns   10

The authors mention on page 9, "When comparing   the NP and SC groups, the NP membranes showed higher tensile strength and   elongation at break values, in addition to lower elastic moduli which were   similar to those of commercial membranes." This is not correct. The   various properties, as clearly evident from Figure 4 show comparable values   for NP and SC groups. In fact, the NP groups with pores (LP and SP) show   lower values for all the three properties. Hence mentioning that the 3D   printed NP group works has superior properties is not correct and must be   modified. Also a clear trend in the various properties is observed as we move   from the LP, SP to NP formulations. It seems quite evident that as the pore   size decreases, the tensile strength and elastic modulus increase, suggesting   a firmer underlying structure. These observations must be explained and   related to the structural integrity and overall morphology of the membrane.

Response 10

The sentence you quoted here is logically correct.   In Figure 4, every property had significant differences between NP and SC   groups. Previously, we only revealed the significant differences in figure   legend of Figure 4 and the results section. Now we have added the signs of   significance in the figure. So, the mechanical properties of NP and SC groups   are not comparable. As for the variation trend in the mechanical properties   of the 3D printed groups, we have added descriptive sentences in results and   discussion sections as you instructed.

Revised Fig. 4 and figure legend:

Figure. 4. Results of the tensile test. (a) Tensile   strength (b) Elastic modulus (c) Elongation at break. (n=5; * p<0.05 compared   to all other groups, # p<0.05 compared to LP and SC groups).

Revised sentences in Tensile test results:

The tensile strengths decreased as the pore size   increased from NP to SP to LP.

The elastic moduli also decreased as the pore size   increased.

Revised sentences in discussion:

When comparing 3D-printed groups with different pore   sizes, NP showed the highest strength and elastic modulus but a low elongation   at break, SP exhibited an ideally moderate modulus, a moderate strength and a   low elongation at break close to NP, and LP was the most ductile, but the   lowest on tensile strength and elastic modulus. These results can be well   expected considering that the materials usually become less “stiff” when the   density decreases as porosity increases.

Major Concerns   11

On page 9, the authors mention "When comparing   the NP and SC groups, the cell growth was largely the same at all time   points...". However, in Figure 6, no images for the SC group membranes   are provided. Kindly provide those SEM images for comparison.

Response 11

The sentence you quoted here was concluded from the   absorbance values in Figure 5. As we can see, no matter on day 1, day 3, or   day 7, NP and SC groups showed very close absorbance values which had no   significant differences with each other. As for Figure 6, the main purpose of   including SEM images here was not to compare the differences in total cell   numbers from SEM alone (because it would be unreliable), but to observe cell growth   distribution on the membranes and find out the explanation for the phenomenon   observed in Figure 5 where the differences of cell numbers grown on different   pore-sized membranes became more and more insignificant as incubation time   increased. Since including images of SC groups here might not provide any   useful information and would take additional space, we considered it   unnecessary. We kindly ask you to reconsider this request.

Major Concerns   12

Sentence restructuring in the conclusion section.   Verb and tense corrections in the text.

Response 12

Although the manuscript had already been proofread   by professionals before submission, now we have revised some sentences for   smoother reading and clearer comprehension for the readers as you instructed.

Revised Conclusions:

In conclusion, membranes fabricated by 3D printing   showed improved mechanical properties compared to those produced via the   conventional solvent casting method. Different pore sizes showed different   advantages in terms of mechanical properties. The effects of the both   variables – fabrication method and pore size - on cell growth were   insignificant. Therefore, 3D printing is proven to be a promising method for   the fabrication of customized barrier membranes used in GBR/GTR. When   possible, the membranes should be designed and 3D printed with specific pore   sizes and materials and in specific forms depending on the clinical condition   of the patient, achieving a more personalized treatment.

Major Concerns   13

Ideally, the membranes made using 3D printing   technology, especially the ones with pores, must be compared with similar   porous membranes made using traditional solvent casting (SC) techniques. I   would recommend adding some control experiments with porous membranes made   using SC or if that is not possible then citing data from previously   published research to draw out conclusions, especially in terms of the   elasticity related properties of these membranes.

Response 13

In fact, in the design phase of this research, we   have considered including porous solvent cast membranes as control groups,   but at last decided not to after consideration. The reason for this is,   fabrication of porous membrane using conventional methods requires a few more   steps on the basis of solvent casting, like salt leaching, phase separation,   or freeze drying. In these processes, additional chemicals and heat   treatments shall be applied, and as a result, the material properties of   control groups will be affected by more variables. For example, the selection   of porogen has significant influences on various properties of porous   material made by solvent casting/salt leaching (Lin, H.R.; Kuo, C.J.; Yang,   C.Y.; Shaw, S.Y.; Wu, Y.J. Preparation of macroporous biodegradable PLGA scaffolds   for cell attachment with the use of mixed salts as porogen additives. J.   Biomed. Mater. Res. 2002, 63, 271–279.). Therefore, we think that it is a   more reliable way to compare the 3D printing and solvent casting methods by   comparing the non-porous samples fabricated by them. And by comparing LP, SP,   and NP to each others, we could find out the trend of the effect pore sizes   had on properties of membranes.

Reviewer 3 Report

In the present manuscript, Zhang and coworkers report a fabrication method for polylactide (PLA) based membrane using 3D printing technique for tissue engineering applications. The pore size of such membranes was varied to study the effect on mechanical properties and cell proliferation using PLA-based scaffolds. The manuscript is well written to most part and the results presented will be of interest to materials science, biomaterials, and tissue engineering community. The reviewer suggests following edits to further improve the quality of current manuscript.

Major Suggestions:

1.      Are there any statistically significant differences between the pore size between LP and SP samples based on microCT data? If not, the results presented here may not represent the true variation between the pore sizes and consequently additional prototypes with significant pore size differences would be needed to accurately study the effect of pore size on PLA scaffolds.

2.      Mechanical properties were not tested using a dog-bone shaped specimen. Please justify the reasoning behind this and if this resulted in premature sample failure at the grips.

3.      Rate of degradation: Given the utility of reported PLA membranes for GBR/GTRs, it is critical to evaluate the rate of degradation. It will provide important perspective to readers to understand if there is any effect on rate of degradation as a function of pore size.

Minor suggestions:

1.      Please include SEM acquisition settings in the methods section (eV, aperture size, detector etc.)

2.      Please include quantified numbers of pore size in the abstract.

3.      Please provide struts size and distance between them as mean values along with standard error/standard deviation to understand data variability.

4.      Absorbance values in figure 5 appears to be within range of 0 to 2. Typically, quantifiable detection limits with reliability for absorbance lies between 0 to 1. Can author please clarify if the data presented in figure 5 has correct absorbance range?

Author Response

Major Suggestions   1

Are there any statistically significant   differences between the pore size between LP and SP samples based on microCT   data? If not, the results presented here may not represent the true variation   between the pore sizes and consequently additional prototypes with   significant pore size differences would be needed to accurately study the   effect of pore size on PLA scaffolds.

Response 1

Was it the porosity in Table.1 you were   mentioning? If it was, then the answer is yes – there were statistically   significant differences between the porosity of LP and SP samples. Thank you   for your kind reminder. Now we have added the signs of significance in   Table.1 and sentences in results section to reveal the significance.

Revised Table. 1:

GroupPorosity (%)LP51.5±8.2 aSP32.5±4.9 bNP1.3±1.0 cSC1.6±2.2 cDifferent superscript     letters indicate significant differences between the groups (p<0.05).

Added sentence:

The porosity of the SP was significantly   higher than those of NP and SC, and LP had the significantly highest   porosity.

Major Suggestions   2

Mechanical   properties were not tested using a dog-bone shaped specimen. Please justify   the reasoning behind this and if this resulted in premature sample failure at   the grips.

Response 2

The   samples used in the tensile test were prepared in the recommended shape and   dimension (specimen type 2) provided in ISO standard 527-3 (Plastics -   Determination of tensile properties - Part 3: Test conditions for films and   sheets). No premature sample failure at the grips was observed during the   test.

Major Suggestions   3

Rate of degradation: Given the utility of reported   PLA membranes for GBR/GTRs, it is critical to evaluate the rate of   degradation. It will provide important perspective to readers to understand   if there is any effect on rate of degradation as a function of pore size.

Response 3

Indeed, the degradation rate is important for   biodegradable materials, but it’s not the aim of this research. As was mentioned   in introduction, the complete degradation of PLA may last for nearly 1 year,   so it should be conducted as a long-term study. We already set up a plan of   long-term degradation study on the degradation rates of our membranes for the   next research topic. Still, thank you for your considerate suggestion.

Minor Suggestions   1

Please include SEM   acquisition settings in the methods section (eV, aperture size,   detector etc.)

Response 1

We have now added   the “accelerating voltage” involved in the methods of SEM taken for sample   morphology and cell proliferation test. But as we know, the other parameters   are generally not mentioned in the papers. We kindly ask you to reconsider   the necessity of including these parameters.

Minor Suggestions   2

Please include   quantified numbers of pore size in the abstract.

Response 2

We have now included   the pore sizes of different groups in the abstract as you instructed.

Revised sentence:

PLA membranes with   three different pore sizes (large pore - 479 μm, small pore - 273 μm, and no   pore) were 3D printed and membranes fabricated using the conventional solvent   casting method were used as the control group.

Minor Suggestions   3

Please provide   struts size and distance between them as mean values along with standard   error/standard deviation to understand data variability.

Response 3

The values of width   of the struts and the distance between struts have been modified as mean   value ± standard deviation as you instructed.

Revised sentences:

For the LP group,   the average width of the struts was 278±58 μm and average distance between   struts was 397±48 μm. For the SP group, the average width of struts was   264±49 μm and average distance between them was 188±43 μm. For the NP group,   the average width of struts was 380±27 μm and no space was observed between   struts except for some defects.

Minor Suggestions   4

Absorbance values   in figure 5 appears to be within range of 0 to 2. Typically, quantifiable detection limits with   reliability for   absorbance lies between 0 to 1. Can author please   clarify if the data presented in figure 5 has correct absorbance range?

Response 4

As is specified in   the manual of the microplate reader used in the test:

accuracy in 0.000-2.000   OD: ± 1% ± 0.010 OD

repeatability in 0.000-2.000   OD: ± 1% ± 0.005 OD.

So we believe the absorbance   values obtained in the test should be reliable enough. We will send the   manual along with this document.

Round 2

Reviewer 2 Report

All questions and concerns were addressed.

Reviewer 3 Report

Authors have addressed all of comments provided by the reviewer.